# Epidemiology and Burden of Sepsis at Thailand’s Largest University-Based National Tertiary Referral Center during 2019

**DOI:** 10.3390/antibiotics11070899

**Published:** 2022-07-05

**Authors:** Lalita Tancharoen, Prat Pairattanakorn, Visanu Thamlikitkul, Nasikarn Angkasekwinai

**Affiliations:** Division of Infectious Diseases and Tropical Medicine, Department of Medicine, Faculty of Medicine Siriraj Hospital, Mahidol University, Bangkok 10700, Thailand; lalita.tan@mahidol.ac.th (L.T.); prat.pai@mahidol.ac.th (P.P.); visanu.tha@mahidol.ac.th (V.T.)

**Keywords:** epidemiology, burden, sepsis, septic shock, blood culture, outcome, mortality

## Abstract

Data specific to the epidemiology and burden of sepsis in low- and middle-income countries are limited. This study aimed to determine the epidemiology and burden of adult patients with sepsis at Siriraj Hospital during 2019. Randomly selected adult patients who had blood cultures performed at our center during January–December 2019 were enrolled. A Quick Sepsis-related Organ Failure Assessment (qSOFA) score was used to determine the presence of sepsis. Demographic data and clinical outcome data were collected, and the annual incidence of sepsis or septic shock and death was estimated. Of the 987 subjects who had blood cultures performed, 798 had infections, 341 had sepsis, and 104 had septic shock. The prevalence of sepsis or septic shock was 34.9% among blood cultured patients, and 42.7% among those with infections. The prevalence of septic shock was 30.5% among subjects with sepsis. Approximately 63% of sepsis subjects were hospital-acquired infections. The factors independently associated with 28-day mortality in sepsis were receiving an immunosuppressive agent (adjusted odds ratio [aOR]: 2.37, 95% confidence interval [CI]: 1.27–4.45; *p* = 0.007), septic shock (aOR: 2.88, 95% CI: 1.71–4.87; *p* < 0.001), and proven infection (aOR: 2.88, 95% CI: 1.55–5.36; *p* = 0.001). Receiving appropriate, definitive antibiotic therapy (ABT) was independently associated with lower mortality in sepsis (aOR: 0.50, 95% CI: 0.27–0.93; *p* = 0.028) and septic shock subjects (aOR: 0.21, 95% CI: 0.06–0.72; *p* = 0.013). Achievement of mean arterial pressure (MAP) ≥ 65 mmHg (aOR: 0.09, 95% CI: 0.01–0.77; *p* = 0.028) and urine output ≥ 0.5 mL/kg/h (aOR: 0.15, 95% CI: 0.04–0.51; *p* = 0.006) were independently associated with lower mortality in septic shock patients. The incidence and mortality of sepsis remains high. Appropriate choice of definitive ABT and achievement of MAP and urine output goals may lower mortality in patients with sepsis or septic shock.

## 1. Background

Sepsis, which is defined as life-threatening organ dysfunction that is caused by dysregulated host response to infection, is a major global health problem [1]. An estimated 48.9 million cases of sepsis were reported worldwide in 2017. Of those, 11 million patients died, accounting for 19.7% of all global deaths in 2017 [2]. Approximately 85% of sepsis cases and sepsis-related deaths occur in low-to-middle-income countries (LMICs) [2]. A recent systematic review reported a sepsis-related hospital mortality rate of 27% [3,4]. In recognition of the threat of sepsis, the World Health Organization (WHO) made sepsis a global health priority. Accordingly, improved prevention, diagnosis, and clinical management of sepsis were included in the agenda of the 70th World Health Assembly (WHA), which was held during May 2017. At that meeting, the WHO urged member states to follow the recommendations set forth in the WHA resolution, including fostering specific epidemiologic surveillance systems, and incorporating the prevention, diagnosis, and treatment of sepsis into national healthcare systems in both community and healthcare settings [5,6]. 

Many tools have been developed to detect sepsis, including the Systemic Inflammatory Response Syndrome (SIRS) criteria in 1991 [7], the Sequential Organ Failure Assessment (SOFA) score in 1994, and the quick Sepsis-related Organ Failure Assessment (qSOFA) score in 2016 [1]. The accuracy of different sepsis scoring systems varies among patient cohorts and settings, and recognition of sepsis using various criteria in different hospital settings remains a diagnostic challenge. A study of the accuracy of various tools to detect sepsis in 460 patients at our center (Siriraj Hospital) showed that a SIRS score ≥ 2, a qSOFA score ≥ 2, and a National Early Warning score (NEWS) ≥ 5 yielded the highest sensitivity (93.2%), specificity (81.3%), and accuracy (72.6%) for detecting sepsis, respectively. Moreover, the positive predictive value of a qSOFA score ≥ 2 for detecting sepsis was 75.5%, which is higher than the positive predictive values of SIRS ≥ 2, SOFA ≥ 2, Modified Early Warning score (MEWS) ≥ 4, and NEWS ≥ 5 for detecting sepsis [8].

The most recent guideline for the management of sepsis recommends a revised ‘hour-1 bundle’ for immediate resuscitation and management of patients with septic shock [9,10]. Although the incidence of sepsis has decreased by 37.0%, and overall sepsis-related mortality has decreased by 52.8% [4], the burden of sepsis remains high in LMICs, including those located in Southeast Asia (SEA). In Thailand, the Ministry of Public Health reported that sepsis-related mortality persisted at an unacceptably high rate of approximately 32% during the years 2017 to 2020 [11,12]. Insufficient critical care capacity was reported to be one of the factors that contributed to the persistently high rate of sepsis-related mortality in SEA [13]. Few studies have investigated and reported the recent epidemiology and burden of sepsis in LMICs [14,15]. Improved understanding of the epidemiology and the burden of sepsis, and the factors independently associated with mortality in sepsis, may facilitate the development and implementation of more efficacious interventions to improve the outcomes of sepsis patients.

## 2. Methods

### 2.1. Study Design and Population

The protocol for this study was approved by the Siriraj Institutional Review Board (SIRB) of the Faculty of Medicine, Siriraj Hospital, Mahidol University, Bangkok, Thailand (COA No. SI 597/2019). Written informed consent was not obtained from subjects since the study involving no more than minimal risk to subjects and all data were anonymized during the study. Patients treated at Siriraj Hospital, which is a 2300-bed university-based national tertiary referral center during January 2019 to December 2019 were enrolled. Eligible subjects were hospitalized adults aged 18 years or older who had a blood culture performed. The primary study outcomes were the prevalence of sepsis among patients who had a blood culture performed, and the 28-day mortality and factors independently associated with mortality among patients with sepsis or septic shock.

### 2.2. Study Procedure

Data specific to patients who had a blood culture performed were retrieved from the hospital database each day during January 2019 to December 2019. Approximately 80–100 adult patients who had blood cultures performed were randomly selected per month using the random generator feature of R program software (The R Project for Statistical Computing, Vienna, Austria). The presence of sepsis according to the qSOFA score was evaluated at the time of blood draw for culture, or within 6 h before blood culture. The worst value for each qSOFA scoring item was used to reach a sepsis determination. The medical records of all included patients were reviewed for demographic data, presence of infection, type and site of infection, causative pathogen, presence of sepsis or septic shock, severity of sepsis, antibiotic and adjunctive therapy, and outcome of sepsis.

### 2.3. Definitions

Patient with infection was defined as a patient who had clinical features of local or systemic infection, such as fever, localized symptoms and signs of infection, and who received antibiotic therapy with or without positive cultures. Proven infection was defined as an infection for which the causative pathogen was identified by culture, antigen or antibody detection, or polymerase chain reaction (PCR) of the specimen taken from the suspected site of infection or blood or histopathologic specimen taken from the suspected site of infection. An automated system with VITEK2 was used for bacterial identification and antimicrobial susceptibility testing (AST). The AST was interpreted by the Clinical and Laboratory Standards Institute (CLSI) 2020 guideline [16]. Community-acquired infection was defined as infection in a patient who was hospitalized <2 days, who had no healthcare-associated conditions, or who was not hospitalized in other hospitals >2 days before transfer to our center. Hospital-acquired infection was defined as an infection in a patient who was hospitalized for >2 days or hospitalized in other hospitals >2 days before transfer to our center, or who had healthcare-associated conditions, including history of prior hospitalization, use of antibiotics within the preceding 90 days, residence in a nursing home or extended care facility, or chronic dialysis within 30 days [17]. Immunosuppressive treatment is determined as receiving corticosteroid >15 mg/day for at least 3 weeks, receiving chemotherapy, or transplant recipients who are receiving immunosuppressive agents. Sepsis was defined as a qSOFA score (range: 0–3) of ≥2 points using the following clinical criteria: one point for low blood pressure (SBP ≤ 100 mmHg), one point for high respiratory rate (≥22 breaths per min), and 1 point for altered mentation (Glasgow Coma Scale < 15) [1]. Septic shock was defined as sepsis with persistent hypotension requiring vasopressor to maintain a mean arterial pressure (MAP) of 65 mmHg or having a serum lactate level > 2 mmol/L (>18 mg/dL) despite receiving adequate volume resuscitation [1]. *Escherichia coli* and *Klebsiella pneumoniae* resistance to ceftriaxone are classified as Extended-spectrum β-lactamase-producing (ESBL) pathogen. *E***.***coli*, *K***.***pneumoniae* and *Acinetobacter baumannii* resistance to carbapenem are classified as Carbapenem-resistant pathogen [18-19]. Empiric antibiotic therapy (ABT) was defined as the antibiotic(s) which was/were given to the subject pending microbiology study results. Concordant empiric ABT was defined as the recovered pathogen being susceptible to the given empiric ABT. Appropriate choice of definitive ABT was defined as the antibiotic which was modified according to the antibiotic susceptibility test results of the isolated causative agent, which had narrower spectrum or was safer than the given empiric ABT. 

### 2.4. Sample Size Estimation

Information from the Global Antimicrobial Resistance Surveillance System (GLASS) at Siriraj Hospital in 2016 showed that the number of patients who had blood culture performed was approximately 5000 patients [20], with a rate of sepsis in patients with positive blood culture specimens of approximately 15%. The present study aimed to include and evaluate a study population amounting to approximately 20% of that 5000-patient value. Therefore, 1000 adult patients who had blood cultures performed during 2019 were randomly selected and included in this study.

### 2.5. Statistical Analysis

The categorical data were reported as frequency and percentage, and continuous variables were reported as mean plus/minus standard deviation for normally distributed variables, and as median and interquartile range (IQR) for non-normally distributed variables. Student’s *t*-test or Mann–Whiney U test were used for the comparison of continuous variables (normally distributed and non-normally distributed, respectively). Chi-square test or Fisher’s exact test was used to compare categorical variables. A *p*-value < 0.05 was considered statistically significant. Variables with a *p*-value < 0.1 from comparative analyses were included in a binary logistic regression analysis to identify factors independently associated with 28-day mortality. All statistical analyses were performed using SPSS Statistics version 20 (SPSS Inc., Chicago, IL, USA). 

## 3. Results

Among 1000 adult patients who had blood cultures performed, 22 patients were excluded due to duplicate data. Out of a total of 978 included subjects with blood cultures performed, 798 subjects had infections. Of those, 341 had sepsis or septic shock, and 104 had septic shock. The prevalence of sepsis or septic shock was 34.9% among patients who had blood cultures performed, and 42.7% among those who had infections. The prevalence of septic shock was 30.5% among patients with sepsis. Demographic and clinical characteristics of 798 patients with infection who had blood cultures performed, and a comparison between those without sepsis and those with sepsis or septic shock, are shown in Table 1. Most subjects (63%) with sepsis or septic shock had hospital-acquired infection, and the most common site of infection was the respiratory tract (43%) followed by the genitourinary tract (13.5%). Bacteremia was observed in 35% of sepsis or septic shock subjects. Twenty-eight-day mortality was much higher in the sepsis or septic shock group compared to the non-sepsis group (37.9% vs. 9.7%, *p* < 0.001). Among the 223 sepsis subjects without shock, 67 (30%) of them died. Sepsis subjects with shock had a significantly higher mortality rate than those without shock (55.6% vs. 30%, *p* < 0.001). The type, frequency, and percentage of 602 causative pathogens detected from 435 patients with infection (54.5%) are shown in Appendix A. A total of 305 patients (70.1%) had infection caused by a single pathogen, 95 patients (21.8%) and 35 patients (8.1%) had mixed infection with two and three pathogens, respectively. Approximately 88% of patients had infection caused by bacteria, mainly Gram-negative bacteria. The four most common organisms detected were *Escherichia coli, Acinetobacter baumannii, Klebsiella pneumoniae,* and *Staphylococcus aureus*. ESBL-producing and Carbapenem-resistant *E.coli* were observed in 33.6% and 2.9% of *E. coli*, respectively. ESBL-producing and Carbapenem-resistant *K.pneumoniae* were observed in 15.1% and 17.8% of *K.pneumoniae*, respectively. Carbapenem-resistant *A. baumannii* was observed in 76.3% of *A.*
*baumannii* and methicillin-resistant *S. aureus* (MRSA) was observed in 15.1% of *S.*
*aureus*. Type of antibiotic prescribed for empiric therapy: (A) 1280 prescription for 771 patients who received empiric mono-antibiotic or combination antibiotic therapy and definitive therapy: (B) 411 prescription for 290 patients who received definitive mono-antibiotic or combination antibiotic therapy.

The characteristics of all patients with sepsis or septic shock, and a comparison between those who survived and those who died within 28 days, are shown in Table 2. Among the 322 sepsis or septic shock subjects with available survival status data, 122 (37.9%) died within 28 days. Sepsis or septic shock subjects who died were more likely to have received immunosuppressive agents, had hospital-acquired infections, had septic shock, had respiratory tract infection, received mechanical ventilation, and received renal replacement therapy, and were less likely to have received an appropriate choice of definitive antibiotic therapy (ABT) than those who survived.

The characteristics of all patients with septic shock, and a comparison between those who survived and those who died within 28 days, are shown in Table 3. Among the 99 septic shock subjects with available survival status data, 55 (55.6%) died within 28 days. Septic shock subjects who died were more likely to have received intravenous (IV) fluid, at least 30 mL/kg in 3 h, and less likely to have received an appropriate choice of definitive ABT, less likely to have achieved mean arterial pressure (MAP) ≥ 65 mmHg, and less likely to have achieved urine output (UOP) ≥ 0.5 mL/kg/h compared to those who survived.

Multivariate analysis for factors independently associated with 28-day mortality in patients with sepsis or septic shock, and in patients with septic shock, is shown in Table 4 and Table 5, respectively. The factors independently associated with 28-day mortality in sepsis or septic shock subjects were receiving immunosuppressive agent (adjusted odd ratio [aOR] 2.37, 95% confidence interval [CI]: 1.27–4.45; *p* = 0.007), septic shock (aOR: 2.88, 95% CI: 1.71–4.87; *p* < 0.001), and proven infection (aOR: 2.88, 95% CI: 1.55–5.36; *p* = 0.001). Receiving an appropriate choice of definitive ABT was independently associated with less mortality in sepsis or septic shock subjects (aOR: 0.50, 95% CI: 0.27–0.93; *p* = 0.028), and in septic shock subjects (aOR: 0.20, 95% CI: 0.06–0.68; *p* = 0.009). Achievement of MAP ≥ 65 mmHg (aOR: 0.09, 95% CI: 0.01–0.77; *p* = 0.028) and achievement of UOP ≥ 0.5 mL/kg/h (aOR: 0.19, 95% CI: 0.04–0.51; *p* = 0.006) were both independently associated with lower mortality in septic shock patients.

Sepsis or septic shock burden relative to morbidity and mortality among adult patients admitted to Siriraj Hospital during 2019 was estimated among 11,700 adult patients who had blood cultures performed during January 2019 to December 2019. Extrapolation of the aforementioned data revealed the following estimates: 4083 patients with sepsis or septic shock, 1244 patients with septic shock, and 1547 sepsis-related deaths during 2019 at Siriraj Hospital.

## 4. Discussion

The present study identified eligible subjects who might have sepsis from those who had blood cultures performed because blood culture is usually recommended in all patients suspected of having sepsis or septic shock, bacteremia, or blood stream infection [21]. This recommendation is consistent with the clinical practice guidelines for diagnosis and treatment of infections at Siriraj Hospital. A qSOFA score of ≥2 was used to determine sepsis in patients with infections who had blood cultures performed because it is a simple method that was reported to have higher specificity and positive predictive value than SIRS ≥ 2, SOFA ≥ 2, MEWS ≥ 4, or NEWS ≥ 5 for detecting sepsis [8]. 

The study hospital is a university-based national tertiary referral hospital that is located in Bangkok. As such, our hospitalized patients usually have severe and/or complicated infections. Many characteristics, including type of infection, site of infection, and causative organisms, which were observed in this 2019 study that included adult patients from all wards of the hospital, including non-medical intensive care units, were similar to those observed in a previous study that included only patients admitted to medical wards of Siriraj Hospital in 2007 [22]. However, the epidemiology of sepsis can vary depending on the subject enrollment criteria and hospital setting. The results of our study revealed bacterial infection (88%) to be the most common cause of infections. However, a 2017 multinational multicenter study among three Southeast Asia countries, of children and adults with community-acquired sepsis, found dengue viruses, *Leptospira* spp., *Rickettsial* spp., *E**. coli*, and influenza viruses were commonly identified causative pathogens [23]. 

The 28-day mortality in sepsis or septic shock subjects observed in the present study during 2019 (37.9%) is higher than the rate found in the 2007 study (34.3%) conducted at our center [22]. The difference between studies could be due to differences in the method and criteria used to enroll subjects between studies. The present study identified sepsis subjects from patients who had blood cultures performed using qSOFA, whereas the previous study enrolled subjects prospectively using SIRS criteria to define sepsis. A recent study to determine the performance of different sepsis detection scoring systems at Siriraj Hospital [8] found SIRS to have higher detection sensitivity; however, qSOFA had lower sensitivity, but higher positive predictive value for detection of sepsis. In the present study, we also observed a difference in patient characteristics compared to the 2007 study. Sepsis subjects in the present study were: older (mean age 66.5 vs. 56.9 years, respectively); more likely to have comorbidities (95% vs. 88.6%), such as diabetes mellitus (33% vs. 19.4%) and chronic kidney diseases (25% vs. 7.5%); more likely to be receiving immunosuppressive agents (17% vs. 2%); and more likely to have a hospital-acquired infection (63% vs. 37.9%), compared to patients in the previous study that was performed in medical patients at the same hospital 12 years earlier [22]. 

The mortality rate among sepsis or septic shock patients in this study was higher than the global sepsis-related mortality rate of 20% during 2017 [4], and higher than the 26.7% rate reported from an updated systematic review of sepsis that was published in 2020 [3]. However, the study of global, regional, and national sepsis incidence and mortality relies mainly on systematic review of literature from high income countries or administrative data and International Classification of Diseases (ICD) data for identifying sepsis and mortality, which unavoidably leads to substantial variation in mortality due to differences in healthcare access, quality index, and the method used for sepsis detection [4]. By way of example, the WHO reported a sepsis-related mortality rate of up to 65% in one region during 2017 [4]. The 28-day mortality rate among septic shock patients in our study was also slightly higher than the rate reported from the aforementioned 2007 study conducted on a different population at our center (55.6% vs. 52.6%, respectively) [22]. This is in contrast to the WHO study on global, regional, and national sepsis incidence and mortality during 1990–2017, which reported a decrease in sepsis-related mortality during the 18-year duration of that study [4]. However, the mortality rate among sepsis patients in the present study cannot be directly compared with the rate from the previous study at the same hospital or with the average global sepsis-related mortality rate due to differences in the methods used to identify sepsis, in the study populations, in the patient characteristics, in the causative pathogens, and in the modes of management.

The mortality rate among patients with septic shock in the present study (55.6%) was higher than the rate reported from a previous systematic review that included 6291 septic shock patients from other countries during 2005 to 2018 (36.7%, 95% CI: 32.8–40.8%) [24]. A retrospective study of 280 septic shock patients in Thailand revealed lower mortality (28.5%) in early septic shock survivors in whom shock was successfully resuscitated and vasopressor was discontinued for 72 h or longer [14]. However, the relatively low mortality in the aforementioned study is partly due to the exclusion of deaths during resuscitation, patients who received palliative care, and patients who were unable to wean off vasopressor for longer than 72 h [14]. Only 12% of septic shock patients in the present study received intravenous fluid at a rate of 30 mL/kg or more in 3 h. Our univariate analysis showed that patients who died were significantly more likely to receive this amount of fluid volume compared to patients who survived; however, and similar to the previous study, no independent association was found in multivariate analysis [25]. Optimal fluid challenge is generally recommended in septic shock patients with hypovolemia [5]. Excess positive fluid balance was found to increase mortality in non-fluid responsive patients resulting from transmural vascular leakage and organ dysfunction [26]. It is possible that most of the septic shock patients in this study were non-fluid responsive; however, this specific information was not available in this study.

This study demonstrates that the burden of sepsis and septic shock in adult patients relative to morbidity (4083 patients) and mortality (1547 deaths) at Siriraj Hospital during 2019 remained high. There are only a few factors that were found to be associated with mortality among sepsis patients that can be modified, including receiving an appropriate choice of definitive ABT, which should be modified according to the antibiotic susceptibility test results of the isolated causative agent, restoration of tissue perfusion, and aiming to achieve MAP of 65 mmHg and UOP ≥ 0.5 mL/kg/h. All three of these factors were found to be independently associated with lower mortality in sepsis shock in both the present study and another previous study [10,25]. Rapid administration of antibiotics was found to be more important than rapid completion of an initial bolus of intravenous fluids for lowering risk-adjusted in-hospital mortality in sepsis patients [25]. In addition, two-thirds of sepsis or septic shock in this study were hospital-acquired. Therefore, more interventions need to be developed and implemented, and existing effective interventions for infection prevention, early detection of sepsis, rapid administration of appropriate antibiotic, and timely appropriate supportive care are urgently needed to decrease the sepsis burden at Siriraj Hospital and to improve the outcomes of sepsis. A surveillance system also needs to be developed and implemented to monitor sepsis-related morbidity and mortality after implementation of the aforementioned additional interventions, and to enforce and monitor the existing effective interventions.

The main strength of this study was that we included adult sepsis patients from almost all departments of our hospital, except the pediatric department. Therefore, the results of this study should be generalizable to different adult patient populations in LMICs. Additional strengths include our use of the qSOFA score to define sepsis status, which is more reliable than retrieving ICD data, and we also collected the types of causative pathogens and the number of isolates for each.

This study also has some mentionable limitations. We enrolled patients who had blood cultures performed as a surrogate for suspected sepsis for initial screening of the enrolled subject; therefore, selection bias cannot be ruled out. Some sepsis or septic shock patients might not have blood cultures performed, or those who received palliative care without specific treatment might not have had blood taken for culture [8]. Such events could have led to an underestimation of hospitalized patients with sepsis. Due to the methods used in this study, some factors might have influenced the opinion of researchers relative to the determination of sepsis, such as known blood culture results, leading to diagnostic bias. Another potential compound limitation is that our data were collected from a single center, and our center is a national tertiary referral center that is routinely referred cases that are complicated and thought not to be treatable at less sophisticated medical centers.

## 5. Conclusions

The burden of sepsis and septic shock relative to incidence, morbidity, and mortality remains high. Approximately one-third of patients who had blood cultures performed had sepsis or septic shock, and Gram-negative bacteria were still the most common pathogens causing sepsis or septic shock, accounting for 61%. Receiving an appropriate choice of definitive ABT and the achievement of MAP and urine output goals may lower mortality in patients with sepsis or septic shock. The data from this study will facilitate the development and implementation of more efficacious interventions to improve the outcomes of sepsis and septic shock patients in Thailand.

## Figures and Tables

**Table 1 antibiotics-11-00899-t001:** Demographics and clinical characteristics of 798 patients with infection who had blood cultures performed, and a comparison between those without sepsis and those with sepsis or septic shock.

Variables	Total	Non-Sepsis	Sepsis or Septic Shock	*p*-Value
	(*n* = 798)	(*n* = 457)	(*n* = 341)	
Age (years), mean (SD)	64.08 (18.3)	62.25 (18.4)	66.53 (17.9)	0.001
Male gender, *n* (%)	401 (50.3)	223 (48.8)	178 (52.2)	0.342
Comorbidities, *n* (%)	752 (94.2)	428 (93.7)	324 (95.0)	0.415
Hypertension	445 (55.8)	253 (55.4)	192 (56.3)	0.791
Diabetes mellitus	271 (34.0)	157 (34.4)	114 (33.4)	0.785
Chronic kidney disease	188 (23.6)	102 (22.3)	86 (25.2)	0.340
Received immunosuppressive agent	139 (17.4)	80 (17.5)	59 (17.3)	0.940
Heart failure	129 (16.2)	73 (16.0)	56 (16.4)	0.865
Hematologic malignancy	116 (14.5)	71 (15.5)	45 (13.2)	0.354
Non-hematologic malignancy	113 (14.2)	64 (14.0)	49 (14.4)	0.884
Autoimmune disease	52 (6.5)	30 (6.6)	22 (6.5)	0.949
Chronic lung disease	44 (5.5)	21 (4.6)	23 (6.7)	0.188
Chronic liver disease	36 (4.5)	18 (3.9)	18 (5.3)	0.367
HIV infection	15 (1.9)	6 (1.3)	9 (2.6)	0.172
Transplant	11 (1.4)	8 (1.8)	3 (0.9)	0.297
Others	209 (26.2)	113 (24.7)	96 (28.2)	0.276
Proven infection, *n* (%)	435 (54.5)	203 (44.4)	232 (68.0)	<0.001
Bacteremia, *n* (%)	163 (20.4)	44 (9.6)	119 (34.9)	<0.001
Type of infection, *n* (%)				
Hospital-acquired	494 (61.9)	279 (61.1)	215 (63.0)	0.565
Community-acquired	304 (38.1)	178 (38.9)	126 (37.0)	0.565
Site of infection, *n* (%)				
Respiratory tract	292 (36.6)	145 (31.7)	147 (43.1)	0.001
Urinary tract	130 (16.3)	84 (18.4)	46 (13.5)	0.064
Gastrointestinal tract	70 (8.8)	30 (6.6)	40 (11.7)	0.011
Primary bacteremia	66 (8.3)	21 (4.6)	45 (13.2)	<0.001
Skin and soft tissue	64 (8.0)	40 (8.8)	24 (7.0)	0.378
Cardiovascular	22 (2.8)	19 (4.2)	3 (0.9)	0.005
Catheter-related BSI	13 (1.6)	7 (1.5)	6 (1.8)	0.801
Central nervous system	10 (1.3)	5 (1.1)	5 (1.5)	0.752
Systemic infection	6 (0.8)	5 (1.1)	1 (0.3)	0.195
Others	27 (3.4)	24 (5.3)	3 (0.9)	0.001
Unknown	131 (16.4)	93 (20.4)	38 (11.1)	0.001
Receiving empiric ABT	771 (96.6)	432 (94.5)	339 (99.4)	<0.001
Concordant empiric ABT	284 (36.8)	129 (29.9)	155 (45.7)	<0.001
Combination empiric ABT	246 (31.9)	107 (24.8)	139 (41.0)	<0.001
Receiving appropriate choice of definitive ABT	290 (36.3)	144 (31.5)	146 (42.8)	0.001
Combination definitive ABT	91 (31.4)	37 (25.7)	54 (37.0)	0.038
Duration of ABT (days), median (IQR)	8.0 (6–14)	9.0 (7.0–14.0)	8.0 (5.5–14.0)	0.173
Discharge status, *n* (%)				
Death within 28 days **	165/767 (21.5)	43/445 (9.7)	122/322 (37.9)	<0.001

Abbreviations: ABT, antibiotic therapy; BSI, blood stream infection; IQR, interquartile range. ** Among 798 subjects with infection, 31 cases (12 in non-sepsis and 19 cases in sepsis or septic shock group) had no data on 28-day survival status because they were transferred to other hospitals or left the hospital against advice.

**Table 2 antibiotics-11-00899-t002:** Characteristics of all patients with sepsis or septic shock, and a comparison between those who survived and those who died within 28 days.

Variables	Total *	Survivor	Death	*p*-Value
	(*n* = 322)	(*n* = 200)	(*n* = 122)	
Age (years), mean (SD)	66.5 (17.9)	66.1 (18.3)	67.7 (17.4)	0.421
Male gender, *n* (%)	166 (51.6)	101 (50.5)	65 (53.3)	0.628
Comorbidities, *n* (%)	306 (95.0)	189 (94.5)	117 (95.9)	0.574
Hypertension	181 (56.2)	120 (60.0)	61 (50.0)	0.079
Diabetes mellitus	108 (33.5)	73 (36.5)	35 (28.7)	0.150
Chronic kidney disease	82 (25.5)	57 (28.5)	25 (20.5)	0.110
Received immunosuppressive agent	58 (18.0)	28 (14.0)	30 (24.6)	0.016
Non-hematologic malignancy	48 (14.9)	27 (13.5)	21 (17.2)	0.364
Hematologic malignancy	44 (13.7)	24 (12.0)	20 (16.4)	0.266
Heart failure	51 (15.8)	31 (15.5)	20 (16.4)	0.831
Chronic lung disease	23 (7.1)	15 (7.5)	8 (6.6)	0.750
Autoimmune disease	22 (6.8)	16 (8.0)	6 (4.9)	0.288
Chronic liver disease	18 (5.6)	11 (5.5)	7 (5.7)	0.928
HIV infection	7 (2.2)	5 (2.5)	2 (1.6)	0.714
Transplant	3 (0.9)	2 (1.0)	1 (0.8)	1.000
Others	89 (27.6)	56 (28.0)	33 (27.0)	0.853
Proven infection, *n* (%)	220 (68.3)	129 (64.5)	91 (74.6)	0.059
Bacteremia, *n* (%)	112 (34.8)	65 (32.5)	47 (38.5)	0.271
Type of infection, *n* (%)				
Hospital-acquired	207 (64.3)	120 (60.0)	87 (71.3)	0.040
Community-acquired	115 (35.7)	80 (40.0)	35 (28.7)	
Site of infection, *n* (%)				
Respiratory tract	142 (44.1)	78 (39.0)	64 (52.5)	0.018
Primary bacteremia	44 (13.7)	23 (11.5)	21 (17.2)	0.148
Gastrointestinal tract	38 (1.8)	24 (12.0)	14 (11.5)	0.887
Urinary tract	43 (13.4)	31 (15.5)	12 (9.8)	0.147
Skin and soft tissue	21 (6.5)	17 (8.5)	4 (3.3)	0.066
Catheter-related BSI	4 (1.2)	3 (1.5)	1 (0.8)	0.593
Cardiovascular	3 (0.9)	2 (1.0)	1 (0.8)	1.000
Central nervous system	4 (1.2)	4 (2.0)	0 (0.0)	0.301
Systemic infection	1 (0.3)	1 (0.5)	0 (0.0)	1.000
Others	3 (0.9)	2 (1.0)	1 (0.8)	0.870
Unknown	34 (10.6)	24 (12.0)	10 (8.2)	0.281
Number of pathogens, *n* (%)				
Single pathogen	149 (67.7)	87 (67.4)	62 (68.1)	0.914
Mixed pathogen	71 (32.3)	42 (32.6)	29 (31.9)	
Septic shock	99 (30.7)	44 (22.0)	55 (45.1)	<0.001
Duration from sepsis to first dose ATB (hour), median (IQR)	0.7 (0.0–2.0)	0.5 (0.0–2.0)	0.73 (0.0–2.0)	0.644
Receiving empiric ABT	320 (99.4)	198 (99.0)	122 (100.0)	0.528
Concordant empiric ABT	150 (46.9)	96 (48.5)	54 (44.3)	0.462
Receiving appropriate choice of definitive ABT	142 (44.1)	99 (49.5)	43 (35.2)	0.012
Duration of ABT (days), median (IQR)	8.0 (5.5–14.0)	10.0 (7.0–15.0)	6.0 (3.0–12.0)	<0.001
Mechanical ventilation	156 (48.4)	84 (42.0)	72 (59.0)	0.003
Renal replacement therapy	55 (17.1)	26 (13.0)	29 (23.8)	0.013
Length of stay (days), median (IQR)	17.0 (1.0–34.5)	22.0 (10.0–46.0)	13.0 (5.0–23.0)	<0.001

* 19 subjects were excluded because they were transferred to other hospitals or left the hospital against advice. Abbreviations: ABT, antibiotic therapy; BSI, blood stream infection; IQR, interquartile range.

**Table 3 antibiotics-11-00899-t003:** Characteristics of all patients with septic shock, and a comparison between those who survived and those who died within 28 days.

Variables	Total *	Survivor	Death	*p*-Value
	(*n* = 99)	(*n* = 44)	(*n* = 55)	
Age (years), mean (SD)	66.5 (17.9)	68.3 (19.0)	66.2 (18.2)	0.585
Male gender, *n* (%)	51 (51.1)	23 (52.1)	28 (50.9)	0.893
Comorbidities, *n* (%)	92 (92.9)	42 (95.5)	50 (90.9)	0.457
Hypertension	54 (54.5)	27 (61.4)	27 (49.1)	0.223
Diabetes mellitus	36 (36.4)	18 (40.9)	18 (32.7)	0.400
Chronic kidney disease	28 (28.3)	16 (36.4)	12 (21.8)	0.110
Received immunosuppressive agent	12 (12.1)	3 (6.8)	9 (16.4)	0.148
Non-hematologic malignancy	17 (17.2)	5 (11.4)	12 (21.8)	0.171
Hematologic malignancy	13 (13.1)	5 (11.4)	8 (14.5)	0.641
Heart failure	21 (21.1)	12 (27.3)	9 (16.4)	0.187
Chronic lung disease	6 (6.1)	4 (9.1)	2 (3.6)	0.402
Autoimmune disease	6 (6.1)	4 (9.1)	2 (3.6)	0.402
Chronic liver disease	7 (7.1)	1 (2.3)	6 (10.9)	0.128
HIV infection	3 (3.0)	2 (4.5)	1 (1.8)	0.583
Transplant	1 (1.0)	1 (2.3)	0 (0.0)	0.444
Others	21 (21.1)	8 (18.2)	13 (23.6)	0.509
Proven infection, *n* (%)	67 (67.7)	29 (65.9)	38 (69.1)	0.737
Bacteremia, *n* (%)	35 (35.4)	14 (31.8)	21 (38.2)	0.510
Type of infection, *n* (%)				
Hospital-acquired	72 (72.7)	32 (72.7)	40 (72.7)	1.000
Community-acquired	27 (27.3)	12 (27.3)	15 (27.3)	
Site of infection, *n* (%)				
Respiratory tract	51 (51.5)	22 (50.0)	29 (52.7)	0.787
Primary bacteremia	13 (13.1)	3 (6.8)	10 (18.2)	0.096
Gastrointestinal tract	11 (11.1)	2 (4.5)	9 (16.4)	0.063
Urinary tract	9 (9.1)	6 (13.6)	3 (5.5)	0.159
Skin and soft tissue	7 (7.1)	6 (13.6)	1 (1.8)	0.042
Catheter-related blood stream infection	2 (2.0)	1 (2.3)	1 (1.8)	1.000
Cardiovascular	1 (1.0)	1 (2.3)	0 (0.0)	0.444
Systemic infection	1 (1.0)	1 (2.3)	0 (0.0)	0.444
Unknown	11 (11.1)	6 (13.6)	5 (9.1)	0.532
Number of pathogens, *n* (%)				
Single pathogen	46 (68.7)	20 (69.0)	26 (68.4)	0.962
Mixed pathogen	21 (31.3)	9 (31.0)	12 (31.6)	
Duration from sepsis to first dose ATB (hour), median (IQR)	0.5 (0–1.9)	0.5 (0–12)	1.0 (0–2.7)	0.177
Receiving empiric ABT	99 (100)	44 (100)	55 (100)	-
Concordant empiric ABT	45 (45.5)	20 (45.5)	25 (45.5)	1.000
Receiving appropriate choice of definitive ABT	34 (34.3)	21 (47.7)	13 (23.6)	0.012
Mechanical ventilation	73 (73.7)	32 (72.7)	41 (74.5)	0.838
Renal replacement therapy	33 (33.3)	10 (22.7)	23 (41.8)	0.045
Fluid resuscitation	73 (73.7)	31 (70.5)	42 (76.4)	0.507
Proportion of patients receiving IV fluid **, *n* (%)	12 (12.1)	2 (4.5)	10 (18.2)	0.039
Initial IV fluid in 3 h (mL/kg), median, (IQR)	7.6 (0–19.8)	7.7 (0–17.5)	9.0 (0–21.0)	0.285
Vasoactive agent	84 (84.8)	36 (81.8)	48 (87.3)	0.452
Received corticosteroid	53 (53.5)	22 (50.0)	31 (56.4)	0.528
Achieve MAP ≥ 65 mmHg, *n* (%) (*n* = 91)	66 (72.5)	37 (97.4)	29 (54.7)	<0.001
Achieve UOP ≥ 0.5 mL/kg/h, *n* (%) (*n* = 88)	36 (40.9)	24 (68.6)	12 (22.6)	<0.001
A decrease in lactate ≥ 10%, *n* (%) (*n* = 73)	28 (38.4)	13 (52.0)	15 (31.2)	0.084

* 5 subjects were excluded because they were transferred to other hospitals or left the hospital against advice. ** Receiving IV fluid (at least 30 mL/kg in 3 h). Abbreviations: ABT, antibiotic therapy; IQR, interquartile range; IV, intravenous; MAP, mean arterial pressure; UOP, urine output.

**Table 4 antibiotics-11-00899-t004:** Multivariate analysis for factors independently associated with 28-day mortality in patients with sepsis or septic shock.

Factors	Crude OR	*p*-Value	Adjusted OR *	*p*-Value
	(95% CI)		(95% CI)	
Hypertension	0.67 (0.42–1.05)	0.080	-	-
Received immunosuppressive agent	2.00 (1.13–3.56)	0.018	2.37 (1.27–4.45)	0.007
Hospital-acquired infection	1.66 (1.02–2.69)	0.041	-	-
Respiratory tract infection	1.73 (1.10–2.72)	0.019	-	-
Skin and soft tissue infection	0.37 (0.12–1.11)	0.076	-	-
Septic shock	2.91 (1.79–4.75)	<0.001	2.88 (1.71–4.87)	<0.001
Proven infection	1.62 (0.98–2.66)	0.06	2.88 (1.55–5.36)	0.001
Receiving appropriate choice of definitive ABT	0.56 (0.35–0.88)	0.013	0.50 (0.27–0.93)	0.028
Mechanical ventilation	1.99 (1.26–3.14)	0.003	-	-
Renal replacement therapy	2.09 (1.16–3.75)	0.014	-	-

Abbreviations: CI, confidence interval; OR, odds ratio; ABT, antibiotic therapy. * Adjusted for hypertension, receiving immunosuppressive agent, hospital-acquired infection, respiratory tract infection, septic shock, proven organism, receiving appropriate choice of definitive ABT, and mechanical ventilation.

**Table 5 antibiotics-11-00899-t005:** Multivariate analysis for factors independently associated with 28-day mortality in patients with septic shock.

Factors	Crude OR	*p*-Value	Adjusted OR *	*p*-Value
	(95% CI)		(95% CI)	
Gastrointestinal tract infection	4.1 (0.84–20.11)	0.081	-	-
Skin and soft tissue infection	0.12 (0.01–1.01)	0.052	-	-
Receiving appropriate choice of definitive ABT	0.34 (0.14–0.80)	0.014	0.20 (0.06–0.68)	0.009
Renal replacement therapy	2.44 (1.01–5.92)	0.048	-	-
Proportion of patients receiving IV fluid	4.67 (0.97–22.55)	0.055	-	-
Achieve MAP ≥ 65 mmHg	0.03 (0.00–0.26)	0.001	0.09 (0.01–0.77)	0.028
Achieve UOP ≥ 0.5 mL/kg/h	0.13 (0.05–0.35)	<0.001	0.19 (0.04–0.51)	0.006
A decrease in lactate ≥ 10%	0.42 (0.16–1.13)	0.087	-	-

Abbreviations: CI, confidence interval; OR, odds ratio; IV, intravenous; MAP, mean arterial pressure; UOP, urine output. * Adjusted for gastrointestinal tract infection, skin and soft tissue infection, receiving appropriate choice of definitive ABT, renal replacement therapy, proportion of patients receiving IV fluid (at least 30 mL/h in 3 h), achieve MAP ≥ 65 mmHg, achieve UO ≥ 0.5 mL/kg/h, and decrease in lactate ≥ 10%.

## Data Availability

The data presented in this study are available on request from the corresponding authors.

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
