# Peer review of "Epidemiology and Burden of Sepsis at Thailand’s Largest University-Based National Tertiary Referral Center during 2019"

_antibiotics, 2022, doi:10.3390/antibiotics11070899_

Round 1
Reviewer 1 Report
The manuscript of Tancharoen et al describes addresses the burden of sepsis on a patients population at a university based hospital in Thailand.
General comments:
The authors provide in the Tables of the manuscript many clinical data. However, no specific data are provided on the pathogens and the antibiotics prescibed. Please provide a Table with indentified microbial pathogens (bacterial and viral species) found in the patients and a possible correlation with mortality. In addition, provide detailed information for the antibiotics that were precribed to the patients.
Specific comments:
- Abstract line 15: delete "42.7%" since percentages for other parameters are not provided.
- Line 15: the authors refer to a group having "sepsis or septic shock". Is it possible to have a septic shock without a sepsis? When not just limit the wording to "sepsis"
- The abstract could be more concise. For example delete lines 26-27. Such an extrapolation could be incorporated in the discussion and does not add any value to the abstract.
- Lines 232 - 233. Please provide a number (or percentage) for the bacterial infections that were found in this study.
- Lines 233 and 234: Leptospira sp., Rickettsial spp and E. coli are also bacteria. Without providing numbers the start of the sentence using "In contrast " is confusing. Please clarify that in the text.
- Line 330: please provide a number (or percentage) when refering to "Gram-negative were still the most common pathogens". From this perspective also a detailed Table including bacterial species found would be very helpful.
Author Response
Response to reviewers' comments
Reviewer 1
The manuscript of Tancharoen et al describes addresses the burden of sepsis on patients population at a university based hospital in Thailand.
General comments:
The authors provide in the Tables of the manuscript many clinical data. However, no specific data are provided on the pathogens and the antibiotics prescribed. Please provide a Table with identified microbial pathogens (bacterial and viral species) found in the patients and a possible correlation with mortality. In addition, provide detailed information for the antibiotics that were prescribed to the patients.
Reply: Thank you the reviewer for the suggestion. The causative pathogens isolated from patients with infection classified by sepsis status are shown in supplementary Table 1 (Table S1). The details of antibiotics that were prescribed as empiric and definitive therapy are shown in supplementary Figure 1A and 1 B (Figure S1). Since there were different types of pathogens causing sepsis in this study, the association of particular pathogen with mortality could not be determined.
Specific comments:
- Abstract line 15: delete "42.7%" since percentages for other parameters are not provided.
Reply: We have deleted the number 42.7% as suggested by reviewer.
- Line 15: the authors refer to a group having "sepsis or septic shock". Is it possible to have a septic shock without a sepsis? When not just limit the wording to "sepsis"
Reply: We have deleted septic shock in line 15.
- The abstract could be more concise. For example delete lines 26-27. Such an extrapolation could be incorporated in the discussion and does not add any value to the abstract.
Reply: We have deleted the lines 26-27 in the abstract as suggested.
- Lines 232 - 233. Please provide a number (or percentage) for the bacterial infections that were found in this study.
Reply: As mentioned earlier in the result section line 159 and more detail in the supplementary Table 1 (Table S1), bacterial infection (87%) was the most common cause of sepsis in this study.
- Lines 233 and 234: Leptospira sp., Rickettsial spp and E. coli are also bacteria. Without providing numbers the start of the sentence using "In contrast " is confusing. Please clarify that in the text.
Reply: We have clarified the sentence as suggested and changed to-However, a 2017 multinational multicenter study among 3 Southeast Asia countries in children and adults with community-acquired sepsis found dengue viruses, Leptospira spp, Rickettsial spp., E. coli, and influenza viruses were commonly identified causative pathogens.
- Line 330: please provide a number (or percentage) when referring to "Gram-negative were still the most common pathogens". From this perspective also a detailed Table including bacterial species found would be very helpful.
Reply: We have added the percentage 61% in line 337. The details of causative pathogen was shown in Table S1.
Reviewer 2 Report
Excellent work to quantify the risk of sepsis/shock in underdeveloped countries and the health care disparities that exist.
Author Response
Excellent work to quantify the risk of sepsis/shock in underdeveloped countries and the health care disparities that exist.
Reply: Thank you.
Reviewer 3 Report
Although the risk profile does not read unexpected, and is therefore rather confirming, this single hospital observation in Thailand has its relevance. This includes the observation that a relevant portion remains ‘hospital acquired’, that ‘proactive’ management has a positive effect on outcome, and that a relevant portion of positive blood cultures were from MDR pathogens.
How were blood cultures collected, managed and analysed ?
How were immunosuppressive treatments defined ?
Was causality assessed, or do the authors report on ‘all group mortality’ ?
Can the authors provide more information on the antibiotic regimens applied ? Was the management at hoc, case based, or based on a protocol ?
Author Response
Although the risk profile does not read unexpected, and is therefore rather confirming, this single hospital observation in Thailand has its relevance. This includes the observation that a relevant portion remains ‘hospital acquired’, that ‘proactive’ management has a positive effect on outcome, and that a relevant portion of positive blood cultures were from MDR pathogens.
How were blood cultures collected, managed and analysed?
Reply: We have randomly selected 1,000 adult patients who had blood cultures performed and excluded 22 patients with duplicate data, 180 patients who have no infection. Therefore, 798 patients with infection were analyzed classifying by status of sepsis as shown in Table1. We made a clarification in line 146-147.
How were immunosuppressive treatments defined?
Reply: Immunosuppressive treatment is determined as receiving corticosteroid > 15 mg/day for at least 3 weeks, receiving chemotherapy and transplant recipients who are receiving immunosuppressive agents. We made a clarification in line 108-110.
Was causality assessed, or do the authors report on ‘all group mortality’ ?
Reply: We reported the association of factors with mortality including demographic data, clinical features and treatment. The causality could not be determined from this study due to the nature of retrospective study.
Can the authors provide more information on the antibiotic regimens applied? Was the management at hoc, case based, or based on a protocol?
Reply: The detail of empiric and definitive antibiotic therapy for infected patient was shown in supplementary Figure 1 (Figure S1: A and B). We added the text in line 168-169. The choice of antibiotics was selected by the responsible physician on a case by case basis.
Round 2
Reviewer 1 Report
The authors improved the manuscript and as required provided data on the identified infective agents and antimicrobial therapy an presented there in Supplementary Table 1 and Supplementary Fig. 1. In addition several textual adaptions were made as required.
Supplementary Table 1 and Supplementary Fig. 1 need improvement.
1. A legend describing data presented in Fig. S1 is missing. This is a requirement. A legend for the X-axis is lacking.
2. Table S1.
a. Mycobacteria are ALSO bacteria and should be listed under "Bacteria". Then the Bacteria make 87 +1.5 = 88.5% of the microorganims with sepsis (incl septic shock). Please also correct this in the main text.
b. Two bacterial species names should be corrected. Correct names: Listeria monocytogenes and Streptococcus pyogenes.
c. What does the category "Other" mean? what kind of microorganisms are these. Or is it simply : "not identified" or "unknown". If these microorganisms are "unknown", one cannot make the conclusion that these are "causative" pathogens as stated in the legend of the table
d. Also the category "other viruses" is very vague. On basis of what is concluded that these patients had a viral infections? Detection methods for virus identification are mostly specific, please specify these viruses. How can it be concluded that these are the causitive pathogen when we have no identification?
e. The calculations of numbers and percentages of microorganisms are not correct. I have discovered many mistakes. For example, it is stated that 604 pathogens were isolated, but in the table I calculate 619, even if subtract the unknowns (n=5), I calculate from the table 614. Secondly, under "sepsis or spetic shock" 294 bacteria are stated. However, when I make an addition of the gram positive (n=82) and and gram negative (n=207) I get the number 289. Also percentages as presented in the figure are not correctly calculated. A complete revision of the numbers presented in Table S1 is required.
In addition modify the legend text of the table: change " causative pathogens isolated" in "causative pathogens isolated or detected". I assume the viruses were not isolated.
Author Response
The authors improved the manuscript and as required provided data on the identified infective agents and antimicrobial therapy an presented there in Supplementary Table 1 and Supplementary Fig. 1. In addition several textual adaptions were made as required.
Supplementary Table 1 and Supplementary Fig. 1 need improvement.
- A legend describing data presented in Fig. S1 is missing. This is a requirement. A legend for the X-axis is lacking.
Reply: We added the label on the axis as suggested.
- 2. Table S1.
- a. Mycobacteria are ALSO bacteria and should be listed under "Bacteria". Then the Bacteria make 87 +1.5 = 88.5% of the microorganims with sepsis (incl septic shock). Please also correct this in the main text.
Reply: Thank you for your useful suggestion. We have listed mycobacteria under bacteria and changed the number in the total of bacteria in Table S1 as suggested.
- b. Two bacterial species names should be corrected. Correct names: Listeria monocytogenes and Streptococcus pyogenes.
Reply: Thank you for pointing out our mistake. We have corrected these two names as suggested.
- c. What does the category "Other" mean? what kind of microorganisms are these. Or is it simply : "not identified" or "unknown". If these microorganisms are "unknown", one cannot make the conclusion that these are "causative" pathogens as stated in the legend of the table
Reply: We made a clarification and have listed all organisms individually. The detail of all causative pathogens were listed in Table S1.
- d. Also the category "other viruses" is very vague. On basis of what is concluded that these patients had a viral infections? Detection methods for virus identification are mostly specific, please specify these viruses. How can it be concluded that these are the causitive pathogen when we have no identification?
Reply: The diagnosis of viral infections was mainly based on molecular identification as we have previously mentioned in the definition part that “the causative pathogen was identified by culture, antigen, antibody detection or PCR of the specimen taken from suspected site of infection or blood or histopathologic specimen.” Among 30 viral infection, 15 influenza/RSV were diagnosed by positive antigen (n=2) or molecular detection by RT-PCR (n=13), 5 dengue virus by NS1 antigen and 4 cytomegalovirus, 2 human metapneumovirus, 2 parainfluenza, 1 adenovirus and 1 chikungunya virus were diagnosed by molecular detection.
- e. The calculations of numbers and percentages of microorganisms are not correct. I have discovered many mistakes. For example, it is stated that 604 pathogens were isolated, but in the table I calculate 619, even if subtract the unknowns (n=5), I calculate from the table 614. Secondly, under "sepsis or spetic shock" 294 bacteria are stated. However, when I make an addition of the gram positive (n=82) and and gram negative (n=207) I get the number 289. Also percentages as presented in the figure are not correctly calculated. A complete revision of the numbers presented in Table S1 is required.
Reply: Thank you for your helpful comments and for taking the time to point out options to improve our manuscript. We have completely revised Table S1 as suggested. We collapsed the column sepsis and non-sepsis into one column since there were a significant number of patients having mixed infection and was difficult to determine which type of pathogen causing sepsis. There were 602 pathogens identified from 435 patients. The cumulative number of pathogen specified in Table S1 would be greater than 435 patients since individual patient may have infected more than one type of pathogen. We made a clarification in results section as follows “The type, frequency, and percentage of 602 causative pathogens detected from 435 patients with infection (54.5%) are shown in Supplementary Table 1. A total of 304 patients (69.9%) had infection caused by single pathogen, 95 patients (21.8%) and 35 patients (8.3%) had mixed infection with two and three pathogens, respectively. Approximately 88% of patients had infection caused by bacteria, mainly Gram-negative bacteria. The four most common organisms detected were…..”
The figure 1 have reported the number of each type of antibiotic used for patients in this study not the percentage as each patient may have received more than one antibiotic. We have labeled the X and Y axis to make a clarification.
In addition modify the legend text of the table: change " causative pathogens isolated" in "causative pathogens isolated or detected". I assume the viruses were not isolated.
Reply: We agree with the reviewer and have changed as suggested.
